# Functional Polymer Nanocomposites with Increased Anticorrosion Properties and Wear Resistance for Water Transport

**DOI:** 10.3390/polym15163449

**Published:** 2023-08-18

**Authors:** Andriy Buketov, Oleksandr Sapronov, Kostyantyn Klevtsov, Boksun Kim

**Affiliations:** 1Department of Transport Technologies and Mechanical Engineering, Kherson State Maritime Academy, Ushakova Avenue, 20, 73003 Kherson, Ukraine; buketov@tntu.edu.ua (A.B.); klevtsovk0226@gmail.com (K.K.); 2School of Engineering, Computing and Mathematics, University of Plymouth, Drake Circus, Plymouth PL4 8AA, UK; boksun.kim@plymouth.ac.uk

**Keywords:** epoxy resin, hardener, nanopowder, filler, corrosion, wear resistance, mechanical characteristics, water transport

## Abstract

Corrosive destruction and hydroabrasive wear is a serious problem in the operation of machine parts and water transport mechanisms. It is promising to develop new composite materials with improved properties to increase the reliability of transport vehicles. In this regard, the use of new polymer-based materials, which are characterized by improved anticorrosion properties and wear resistance, is promising. In this work, therefore, for the formation of multifunctional protective coatings, epoxy dian oligomer brand ED-20, polyethylene polyamine (PEPA) hardener, a mixture of nanodispersed compounds with a dispersion of 30–90 nm, fillers Agocel S-2000 and Waltrop with a dispersion of 8–12 μm, and particles of iron slag with a dispersion of 60–63 μm are used for the formation of multifunctional protective coatings. Using the method of mathematically planning the experiment, the content of additives of different physico-chemical natures in the epoxy binder is optimized to obtain fireproof coatings with improved operational characteristics. A mathematical model is developed for optimizing the content of components in the formation of protective anticorrosion and wear-resistant coatings for means of transport as a result of the complex effect of a mixture of nanodispersed compounds, iron scale, and Waltrop. Based on the mathematical planning of the experiment, new regularities of increasing the corrosion resistance and resources of the means of transport are established through the formation of four different protective coatings, which are tested for resistance to aggressive environments (technical water—CAS No. 7732-18-5, gasoline—CAS No. 64742-82-1, acetone—CAS No. 67-64-1, I-20A lubricant—CAS No. 64742-62-7, sodium solutions—CAS No. 1310-73-2, and sulfuric acid—CAS No. 7664-93-9) and hydroabrasive wear resistances. A study of the change in the permeability index in aggressive environments is additionally carried out, taking into account the rational ratio of dispersive fillers in the epoxy binder, which made it possible to create an effective barrier to the penetration of aggressive water molecules into the base. A decrease in the permeability of protective coatings by 2.0–3.3 times relative to the epoxy matrix is achieved. In addition, the wear resistance of the developed materials under the action of hydroabrasion is investigated. The relative resistance of the CM to the action of hydroabrasion was found by the method of materials and coatings testing on the gas-abrasive wear with a centrifugal accelerator. This method enables one to model the real process of the wear of mechanism parts under the hydroabrasive action. It is shown that the coefficient of the wear resistance of the developed materials is 1.3 times higher than that of the polymer matrix, which indicates the resistance of the composites to the influence of hydroabrasive environment. As a result, modified epoxy composite protective coatings with improved anticorrosion properties and wear resistance under hydroabrasive conditions are developed. It is established that the protective coating filled with particles of a mixture of nanodispersed compounds (30–90 nm), iron scale (60–63 μm), and Waltrop (8–12 μm) has the lowest permeability indicators. The permeability in natural conditions of such a coating during the time *t* = 300 days of the study is *χ* = 0.5%, which is 3.6 times less than the similar indicators of the epoxy matrix. It is substantiated that the protective coating filled with particles of a mixture of nanodispersed compounds (30–90 nm), iron scale (60–63 μm), and Agocel S-2000 (8–12 μm) is characterized by the highest indicators of wear resistance. The coefficient of wear resistance under the action of hydroabrasion of such a coating is *K* = 1.75, which is 1.3 times higher than the similar indicators of the original epoxy matrix.

## 1. Introduction

The development of transport technologies at the current stage creates conditions for improving not only the logistics of international transportation but also the development of new materials with improved properties to increase the reliability of transport means [1,2,3,4,5,6,7,8].

It should be noted that modern scientific and technical progress is impossible without the creation of new special epoxy materials that have improved properties [9,10,11,12,13]. In particular, these are stiffness and elasticity, hardness and superfluidity, solubility in water and gas permeability, heat resistance and dielectric properties, and corrosion and wear resistance. Such opposite properties, which exclude each other, are combined in epoxy polymers. At the same time, today, epoxy polymers are widely used in the operation of water transport because of their unique consumer properties.

To improve the properties of polymers, including those based on epoxy, plasticizers and fillers are introduced into the oligomer [14,15,16,17,18,19,20,21]. It should be noted that the introduction of modifiers is promising, since such additives with a small content (1–5%) allow a significant increase in the indicators of adhesive and cohesive strength of epoxy composites [10,13,20,21,22,23,24,25,26,27,28,29]. The latter, in turn, should be used in the form of matrices for protective coatings of working parts, including during their repair in voyage conditions of sea and river transport vessels.

Taking into account the above, researchers [15,16,17,18,30,31,32,33,34,35] showed that with the development of the transport industry, the efficiency of the restoration of parts and mechanisms is important, while increasing their service life and maintainability is relevant. It is necessary in this direction to use new materials that differ not only in improved properties but also in the economic efficiency in their application [30,31]. Anticorrosion protection of parts of river and sea transport is also important, as it provides for an increase in the inter-repair resources of their operation—in this regard, the use of nanofillers.

In addition, today, more and more attention is paid to the secondary processing of metallurgical production waste, which allows, in addition to cleaning the environment, the saving of significant funds aimed at the disposal of the same waste [29,36]. Therefore, it is proposed in our work to use waste of different compositions as fillers for the formation of epoxy composites with increased operational characteristics. The use of cheap fillers (waste from industrial production), which in addition to improving the operational characteristics of epoxy composites contribute to reducing the cost of structural materials, is beneficial from an economic point of view. It is interesting to use iron slag (IS) particles as a filler characterized by the following composition, %: SiO_2_—0.803; Al_2_O_3_—0.36; Fe_2_O_3_—32.57; FeO—64.85; MgO—0.03; MnO—0.44; CaO—0.21; S—0.031. Iron slag is a waste in the production of steel at metallurgical enterprises.

The analysis of the direction of publications researching the properties of epoxy CMs indicates the need to introduce synthesized oxide or carbide fillers into the oligomer to obtain materials with increased operational characteristics. It is proven [14,15,16,17,18,19,20,21,22] that the introduction of dispersed particles into the resin, even in small quantities, ensures a change in the speed of the course of the physicochemical processes during the structure formation of CM. Moreover, not only the chemical and physical nature but also the content and dispersion of the particles are important in regulating the crosslinking processes of epoxy CMs.

The development of coatings with synthesized microdispersed fillers can be interesting for industrial customers. This allows not only the improvement of the properties of coatings but also a significant reduction in their costs. In addition, this approach simultaneously ensures the cost effectiveness of the developed materials and is beneficial from the point of view of ecology and environmental protection, which can be significant to the industry and the development of any region as a whole.

Researchers [37,38] formed ultrahard microdispersed powders by high-voltage electric discharge (VED) synthesis. The research results showed that, as a result of the VED treatment, all the treated particles were crushed and their phase composition changed with the synthesis of high-modulus TiC and Fe_3_C compounds. Similarly, the same authors synthesized a mixture of powders from a charge of Al (15%) + Ti (85%) with the following composition: Ti (75%) + Al_3_Ti (15%) + Ti_3_AlC_2_ (10%). 

During our work, a nanodispersed filler is introduced into the binder to improve the properties of epoxy composites. The powder synthesized by us according to the technology described in the work [38] is used as a nanodispersed filler for experimental studies. Nanopowder is a mixture of nanodispersed compounds (MNDC) and is characterized by the following composition, %: Si_3_N_4_—90, I_2_O_3_—5; Al_2_O_3_—5. Particle size is *d* = 30–90 nm.

With the use of these powders, the authors [37,38] developed new epoxy composite coatings to improve the operational characteristics of the technological equipment. At the same time, it should be noted that the problem of improving the anticorrosion properties and wear resistance of CM in the complex has not yet been finally resolved. The field of polymer materials science, which covers the substantiation of the mechanism of interaction of composition ingredients during their structuring, requires further fundamental research. It is precisely such processes during the formation of the structure of materials that further determine their cohesive properties and, as a result, corrosion and wear resistance.

It should be noted that, for this purpose, various methods of studying the properties of materials are used in the work. On the one hand, there are methods for studying the physical and mechanical properties of materials: modulus of elasticity when bending (ISO 178:2010) and impact resilience (ISO 179-1). On the other hand, a set of corrosion resistance research methods are also used:Corrosive resistance of the protective coatings is determined by immersion of specimens in water and 10% sulfuric acid solution. The duration of exposition specimens with a size of 60 × 10 × 10 mm in aggressive media is 150 days at temperature *T* = 20 ± 2 °C.The corrosion resistance of CM is also determined by immersing the samples in technical water (CAS No. 7732-18-5), gasoline (CAS No. 64742-82-1), acetone (CAS No. 67-64-1), I-20A lubricant (CAS No. 64742-62-7), solutions of sodium (CAS No. 1310-73-2), and sulfuric acid (CAS No. 7664-93-9). The duration of exposure of samples with a size of 60 × 10 × 10 mm in aggressive environments is 720 h at a temperature of *T* = 293 ± 2 K.

Wear resistance according to ISO 9352 is additionally investigated in the work. The relative resistance of the CM to the action of hydroabrasion was found by the method of materials and coatings testing on the gas-abrasive wear with a centrifugal accelerator. 

In our opinion, only such a comprehensive approach to evaluating properties may allow the choosing of a material that will be characterized by maximum durability during operation on water transport.

Taking this into account, the introduction of reactive microdispersed fillers, including synthesized nano- and microdispersed powders, into the epoxy resin is relevant. Chemically active microdispersed fillers in the form of Agocel S-2000 (AC) (Mineral Building Solutions, Germany) and Waltrop (BT) (Waltrop, Germany) powders are used in the formation of materials during our work. These are white and yellow powders, respectively, produced in Germany. They are used as fillers for functional-purpose materials. The size of the particles is 8–12 μm. It is important to carry out research on determining the critical content of powders, which are waste products of industrial production, for the purpose of forming protective epoxy coatings for parts of technological equipment. In addition, the combination of these fillers in the composite will be expedient and effective. This will allow the obtaining of a synergistic effect in improving their properties.

Based on the above, the authors assumed that the noted requirements are largely satisfied when using a mixture of nanodispersed compounds, synthesized microdispersed powders of Agocel S-2000 and Waltrop, and production waste in the form of iron slag particles as filler. Such additives are sufficiently active to accelerate the processes of interphase structure formation relative to the original epoxy matrix.

The purpose of this study is to establish the regularities of the influence of the optimal content of nano- and microdispersed filler based on a mixture of nanodispersed compounds, synthesized microdispersed powders of Agocel S-2000 and Waltrop, and production waste in the form of iron slag particles on the main mechanical, corrosion properties, and wear resistance of polymeric composites from the epoxy matrix.

At the same time, to achieve the goal, the authors plan to:As a result of the mathematical planning of the experiment, optimize the content of fillers to obtain epoxy composites with increased operational characteristics and protective coatings based on them, which work under the influence of hydroabrasives and aggressive environments;Investigate the corrosion resistance of the developed materials in various aggressive environments;Investigate the resistance to hydroabrasive action of the developed materials;Issue recommendations on the creation of protective coatings based on the developed composites for their introduction into the industry.

## 2. Materials and Methods

### 2.1. Materials

Epoxy dian oligomer brand ED-20 (ISO 18280:2010, Technobudresurs, Kyiv, Ukraine) [14,15].

Polyethylene polyamine (PEPA) hardener (TU 6-05-241-202-78, Technobudresurs, Kyiv, Ukraine) [25].

The powder synthesized according to the technology described in the work [38] is used as a nanodispersed filler for experimental studies. Nanopowder is a mixture of nanodispersed compounds (MNDC), %: Si_3_N_4_—90, I_2_O_3_—5, and Al_2_O_3_—5 (Table 1). Particle size is *d* = 30–90 nm.

The optimal filler (MNDC) content is—*q* = 0.25–0.50 pts.wt. At the same time, the destructive bending stresses increase from *σ_fl_* = 48.0 MPa (for the unfilled matrix) to *σ_fl_* = 58.0 MPa, the modulus of elasticity at bending from *E* = 2.8 GPa (for the unfilled matrix) to *E* = 4.0 GPa, and impact resilience reduces from *W* = 7.4 kJ/m^2^ (for unfilled matrix) to *W* = 7.3 kJ/m^2^.

Particles of Agocel S-2000 (AC) (Mineral Building Solutions, Germany) and Waltrop (BT) (Waltrop, Germany) powders are used as microdispersed fillers for experimental studies (Table 2). These are white and yellow powders, respectively, produced in Germany. They are used as fillers for CM for construction purposes. The sizes of the particles are 8–12 μm.

Additionally, iron slag particles (IS) are used as a filler, characterized by the following composition, %: FeO—64.85; SiO_2_—0.803; Al_2_O_3_—0.36; Fe_2_O_3_—32.57. Iron slag is a waste in the production of steel at metallurgical enterprises. Particle size is *d* = 60–63 μm.

### 2.2. Research Methods

The elastic modulus was determined according to ISO 178:2010. Sample parameters are specified in the work [29]. The modulus of elasticity at axial deflection (longitudinal elasticity) is studied. Research is carried out according to the scheme of the 4th bend. The test speed is 0.5–1.0 mm/min according to ISO 178:2010.

The impact resilience was determined according to ISO 179-1 on the MK-30 rebound pendulum machine [10]. Sample parameters are (63 × 12 × 12) ± 0.5 mm.

Corrosive resistance of the protective coatings is determined by the immersion of specimens in water and 10% sulfuric acid solution. The duration of exposing the specimens with a size of 60 × 10 × 10 mm in aggressive media is 150 days at temperature *T* = 20 ± 2 °C. Specimens with a mass of 10.0–12.0 g prior to study and after exposure to aggressive media are weighed on an electronic scale, a DRS-8000 «Shimadzu» with an accuracy of 0.02 ± 0.001 g.

The calculation of the penetration of aggressive media was carried out according to the following formula [28].

The corrosion resistance of polymers was also determined by immersing samples in technical water (CAS No. 7732-18-5), gasoline (CAS No. 64742-82-1), acetone (CAS No. 67-64-1), I-20A lubricant (CAS No. 64742-62-7), solutions of sodium (CAS No. 1310-73-2), and sulfuric acid (CAS No. 7664-93-9).

The duration of the exposure of samples with dimensions of 60 × 10 × 10 mm in aggressive environments was 720 h at a temperature of *T* = 293 ± 2 K. Samples with a mass of 1.0–1.2 g before research and after exposure to aggressive environments are weighed on analytical balances of the VLR-200 brand with an accuracy of up to ±0.001 g.

Wear resistance of the sample was investigated according to ISO 9352. Sample parameters were (100 × 100 × 3) ± 0.5 mm. The sliding speed during wear was 0.28 m/s. When evaluating the wear, the mass of the worn material was taken into account at the standard length of the friction path.

The relative resistance of the polymers to the action of hydroabrasion was found by the method of materials and coatings testing on the gas-abrasive wear with a centrifugal accelerator. The method of conducting the experiment is given in detail in work [23]. The relative intensity of wear is given by the following formula:$$I=\frac{m_{0}-m_{end}}{m_{0}}\cdot 100\%,$$
where m_0_ and m_end_, kg, are the masses of the specimen at the beginning and at the end of the tests, respectively.

The coefficient of wear resistance is determined by the formula:$$K=\frac{I_{E}}{I},\;$$
where *I_E_*—is the relative wear intensity of the standard (St 3 steel) %; *I*—is the relative intensity of CM wear, %.

The structure of the CM after friction was investigated on a XJL-17AT metallographic microscope, which was equipped with a Levenhuk C310 NG (3.2 MegaPixels) camera (Levenhuk, CA, USA).

The structure and microrelief of the surfaces are studied using a LEO EVO 50 scanning electron microscope (Zeiss, Cologne, Germany).

While investigating the physical and mechanical properties of the epoxy material, the deviation of values within 5% of the nominal value was observed.

While investigating the corrosion resistance and hydroabrasive wear resistance of the epoxy material, the deviation of values within 4–6% of the nominal value was observed.

Statistical methods: the statistical processing of the obtained results was performed using the software package for statistical data processing: Graph Pud Prism 8 (GraphPad Software 2365 Northside Dr. Suite 560 San Diego, CA, USA), with the definition of Student’s criterion.

### 2.3. Material-Forming Technology

Epoxy composites were formed using the following technology: first, heating the resin to a temperature of *T* = 353 ± 2 K and then holding it at this temperature for a time of *τ* = 20 ± 0.1 min; the hydrodynamic combination of oligomer and modifier over time *τ* = 1 ± 0.1 min; ultrasonic treatment (UST) of the composition during the time *τ* = 1.5 ± 0.1 min; cooling the composition to room temperature over time *τ* = 60 ± 5 min; introduction of hardener and mixing of the composition over time τ = 5 ± 0.1 min.

The CM was hardened according to the following regime [10,14,23].

## 3. Results and Discussion

### 3.1. Mathematical Planning of the Experiment

During the operation of coating intended for the protection of ship hulls, deck mechanisms, and superstructures, the ability to maintain cohesive strength under the influence of natural external factors is important. At the same time, the development of the surface layer of the coating with high physical and mechanical properties (bending modulus and impact resilience) allows the counteracting of the penetration of an aggressive environment through the polymer to the base. Therefore, the optimal content of fillers Waltrop, Agocel S-2000, and iron slag was determined in previous works. It was established that the improvement of the properties of composite materials was observed when Waltrop was introduced at a content of *q* = 10 pts.wt. At the same time, the destructive bending stresses increase from *σ_fl_* = 48.0 MPa (for the unfilled matrix) to *σ_fl_* = 45.2 MPa, the modulus of elasticity at bending from *E* = 2.8 GPa (for the unfilled matrix) to *E* = 4.0 GPa, and impact resilience from *W* = 7.4 kJ/m^2^ (for unfilled matrix) to *W* = 7.9 kJ/m^2^. The optimal content of Agocel S-2000 filler is *q* = 30 pts.wt. At the same time, the destructive stresses during bending are *σ_fl_* = 38.2 MPa, the modulus of elasticity during bending is *E* = 4.3 GPa, and the impact resilience is *W* = 7.7 kJ/m^2^. The optimal content of iron slag filler is *q* = 60 pts.wt. At the same time, the destructive stresses during bending are *σ_fl_* = 54.6 MPa, the modulus of elasticity during bending is *E* = 4.4 GPa, and the impact resilience is *W* = 8.2 kJ/m^2^.

However, the formation of composites with a two-component filler is interesting from a practical point of view, as it, in our opinion, will allow the improvement of the properties of the studied CMs in the complex. In this context, it is expedient and necessary to use the method of mathematically planning the experiment, which will allow the reduction in the number of conducted studies and the optimization of the content of ingredients to obtain CM with the maximum indicators of the selected characteristics.

On the basis of the above, iron slag and Waltrop were used for mathematically planning the experiment. The dispersion of particles according to granulometric analysis is as follows: iron scale is 60–63 μm and Waltrop is 8–10 μm. For standardization, as well as to simplify calculations, each component (filler) was coded with conventional units taking into account the variation step (Table 3).

Nine experiments (N = 9) were conducted, each of which was repeated three times (*p* = 3) to exclude systematic errors (Table 4).

In order to get an orthogonal planning matrix [5], the adjusted values of the level $x^{\prime }$ are introduced:(1)$$x_{\prime }^{i}={\left(x_{i}\right)}^{2}-\frac{\sum_{u-1}^{N}x_{iu}^{2}}{N}$$

The extended planning matrix of the full factorial experiment (FFE) is shown in Table 5.

The mathematical model y = f (x_1_, x_2_) is devised in the form of a regression equation:(2)$$y=b_{0}^{}+b_{1}x_{1}+b_{2}x_{2}+b_{11}x_{1}^{2}+b_{22}x_{2}^{2}+b_{12}x_{1}x_{2}$$

Regression coefficients are determined using the following formula:(3)$$b_{i}=\frac{\sum_{u=1}^{N}x_{i}y_{i}}{\sum_{u-1}^{N}x_{iu}^{2}}$$

The obtained coefficients of the regression equation are shown in Table 6.

The following regression equation is obtained when analyzing the modulus of elasticity during bending:$$y=5.06-0.63x_{1}-0.12x_{2}-0.03x_{1}^{2}-0.08x_{2}^{2}+0.18x_{1}x_{2}$$

For the statistical processing of the obtained results of the experiment, a check of the reproducibility of the experiments is carried out according to Cochrane’s test:(4)$$G=\frac{S_{u}^{2}{}_{\max}}{\sum_{u=1}^{N}S_{u}^{2}}\leq G_{\left(0.05;\;f_{1};\;f_{2}\right)}\;$$
where S^2^_ui_ is the dispersion that characterizes the results of experiments on the *i*-th combination of factor levels for n = 3; n is the number of parallel experiments; S^2^_u max_ is the largest of the dispersions in the rows of the plan.

Adequacy dispersion is determined by the following formula:(5)$$S_{ui}^{2}=\frac{\sum_{i=1}^{m}{\left(y_{i}-\bar{y_{i}}\right)}^{2}}{m-1}$$
where y_im_ is the value obtained from each parallel experiment; y_i_ is the average value obtained during parallel experiments.

The reproduction dispersion is determined by the following formula:(6)$$\sigma^{2}\left\{y\right\}=\frac{\sum_{i=1}^{N=9}\sigma^{2}{\left\{y\right\}}_{i}}{N(m-1)}$$
where $\sigma^{2}{\left\{y\right\}}_{i}=\sum_{i=1}^{m=3}{\left(y_{i}-\bar{y_{i}}\right)}^{2}$
(7)$$\sigma_{}^{2}\left\{y\right\}=\frac{a^{2}\left\{y\right\}}{N},\;\mathrm{or}\;S_{b_{0}}^{2}=\frac{S_{0}^{2}}{N}$$

The values of dispersions are given in Table 7.

At the same time:$$\sum_{i=1}^{N}S_{ui}^{2}=0.15$$
$$\sigma^{2}\left\{y\right\}=S_{0}^{2}=0.017$$

The calculated values of Cochrane’s test at 5% equal significance is:(8)$$G=\frac{S_{u_{\max}}^{2}}{\sum_{i=1}^{N}S_{ui}^{2}}\;$$
$$G=\frac{0.03}{0.15}=0.067.$$

Checking the results of the experiment according to Cochrane’s test [5] for a fixed probability α = 0.05 confirmed the reproducibility of the experiments. The dispersion characterizes the results of experiments on the i-th combination of factor levels: $S_{u\max}^{2}=0.03$. The calculated value of Cochrane’s test is *G* = 0.067.

The table value of Cochrane’s test is *G_t_* = 0.478.

Therefore, condition (7) is fulfilled:$$G=0.067\leq G_{t}=0.478\;$$

Furthermore, the significance of the coefficients of the regression equation is determined by analyzing the results according to the experimental plan (Table 8).

Furthermore, the dispersion of the regression coefficients (Table 9) are determined according to the following formula:(9)$$S_{b_{i}}^{2}=\frac{S_{0}^{2}}{\sum_{u-1}^{N}x_{iu}^{2}}$$

The significance of the regression coefficients is determined by Student’s test [5]. At the same time, the tabular (*t_t_*) and calculated criterion (*t_c_*) of Student’s test are determined (Table 9).

Depending on the degrees of freedom, *f* = N (n − 1) = 9 (3 − 1) = 18, the tabular value of Student’s test is determined, which is *t_t_* = 2.1.

The calculated values of Student’s test (*t_c_*) and the significance of the coefficients are determined: *t*_0*c*_*, t*_1*c*_*, t*_2*c*_*, t*_11*c*_*, t*_22*c*_*, t*_12*c*_
*> t_t_.*

At the same time:(10)$$t_{ip}=\frac{\left|b_{i}\right|}{S_{b_{i}}^{}};$$

The calculated values of Student’s test, *t*_0*c*_, *t*_1*c*_, *t*_2*c*_, and *t*_12*c*_, are greater than *t_t_*, so it is considered that the coefficients of the regression equation, *b*_0_, *b*_1_, *b*_2_, and *b*_12_, are significant. The calculated values of *t*_11*c*_ and *t*_22*c*_ are smaller than *t_T_*, so the coefficients *b*_11_ and *b*_22_ are not significant. As a result of discarding insignificant coefficients, the following regression equation is obtained:$$y=5.06-0.63x_{1}-0.12x_{2}+0.18x_{1}x_{2}$$

The adequacy of the obtained model is checked according to Fisher’s test [39]:(11)$$F_{p}=\frac{S_{u\max}^{2}}{S_{y}^{2}}\leq F_{\left(0.05;\;f_{{}_{a\sigma }}^{};\;f_{y}\right)}$$
where $S_{u}^{2}{}_{\max}=0.03$ is the calculated value of the dispersion of adequacy (Table 7):(12)$$S_{y}^{2}=\frac{\sum_{i=1}^{N}S_{ui}^{2}}{N};$$

*S_y_^2^* = 0.017 is the dispersion of reproduction.

Then, *F_c_* = 0.6.

$F_{\left(0.05;\;f_{a\sigma };\;f_{u}\right)}$ is the tabular value of Fisher’s test at the 5% level of significance (*f*_1_ = N − (*k* + 1) = 9 − (4 + 1) = 4, *f_2_* = N (n − 1) = 9 (3 − 1) = 18). Then, *F_(t)_* = 2.93 [39].

The calculated value of Fisher’s test is smaller than the table value; therefore, condition (10) is fulfilled. It can be assumed that the equation adequately describes the composition formulation.

The process of interpreting the obtained mathematical model, as a rule, is not reduced only to determining the influence of factors. A simple comparison of the absolute value of the linear coefficients does not determine the relative degree of influence of the factors, since there are also quadratic terms and pairwise interactions. In the detailed analysis of the obtained adequate model, it is necessary to take into account that, for the quadratic model, the degree of influence of the factor on the change of the initial value is not constant.

Dependencies connecting normalized and natural values of variable factors have the following form:(13)$$x_{i}=\frac{q_{i}-q_{i0}}{\Delta q_{i}}$$
where *q_i_* is the value of the *i*-th factor of the experiment, *q_i0_* is the zero level value, and *Δq_i_* is the variation interval [5].

Substituting these values according to Equation (13) into the regression equation and transforming it, we obtain the following regression equation with the natural value of the variable parameters:$$E=12.23-0.099q_{1}-0.138q_{2}+0.0018q_{1}q_{2}\;$$

The given equation in natural values allows us to only predict the value of the initial value for any point in the middle of the factor variation area. However, with its help, it is possible to plot graphs of the dependence of the initial value (modulus of elasticity during bending of composites) on any factor (or two factors). The geometric interpretation of the response surface is shown in Figure 1, Figure 2 and Figure 3.

Based on experimental studies, it is established that both factors are significant. It should be noted that the influence of the content of the main filler on the indicators of the elastic modulus during bending is higher compared to the additional one (according to the Pareto map). Analyzing the calculated response surface, it is determined that the developed epoxy composite with a two-component polydispersed filler with the following particle content, iron slag—60–70 wt% and Waltrop—10–20 wt%, has the optimal indicators of the modulus of elasticity during bending (*E* = 5.5–5.8 GPa).

Similar to the above scheme of calculations, the composition of CM is optimized according to the impact viscosity indicators. The coding of the natural values of the components and the scheme of planning the experiment are selected according to Table 3 and Table 4, respectively.

In the process of analyzing the results of the study of the impact resilience of composites, regression coefficients are obtained and presented in Table 10.

As a result, the following regression equation is derived:$$y=8.36-0.50x_{1}-0.02x_{2}-0.23x_{1}^{2}-0.28x_{2}^{2}+0.13x_{1}x_{2}$$

For the statistical processing of the obtained results of the experiment, the reproducibility of the experiments is checked according to Cochrane’s test [5].

The values of dispersions, which are determined by Equations (5) to (7), are given in Table 11.

At the same time:$$\sum_{i=1}^{N}S_{ui}^{2}=0.31,\;\sigma^{2}\left\{y\right\}=S_{0}^{2}=0.034$$

Then, the calculated value of Cochrane’s test at the 5% level of significance is determined by Equation (8):$$G_{p}=\frac{0.07}{0.31}=0.226\;$$

Checking the results of the experiment according to Cochrane’s test [5] for a fixed probability, α = 0.05, confirmed the reproducibility of the experiments. Dispersion, which characterizes the results of experiments on the *i*-th combination of factor levels, is $S_{u\max}^{2}=0.07$. The calculated value of Cochrane’s test is *G* = 0.226.

The table value of Cochrane’s test is *G_t_* = 0.478.

Therefore, the condition is fulfilled:$$G=0.226\leq G_{t}=0.478\;$$

At the next stage, the significance of the coefficients of the regression equation is determined, analyzing the results according to the experimental plan, and presented in Table 12.

Equations (9) and (10): the significance of the regression coefficients is determined by Student’s test, the tabular value of which is *t_t_* = 2.1 [39]. The calculated values of Student’s test are given in Table 13.

The calculated values of Student’s test, *t*_0*c*_, *t*_1*c*_, and *t*_2*c*_, are greater than *t_t_*, so it is considered that the coefficients *b*_0_, *b*_1_, and *b*_2_ of the regression equation are significant. The calculated values of *t*_11*c*_, *t*_22*c*_, and *t*_12*c*_ are smaller than *t_t_*, so the coefficients *b*_11_, *b*_22_, and *b*_12_ are not significant. As a result, the following regression equation is obtained:$$y=8.36-0.50x_{1}-0.28x_{2}^{2}\;$$

The adequacy of the obtained model is checked according to Fisher’s test [5].

The calculated value of the dispersion of adequacy is $S_{u\max}^{2}=0.07$ (Table 11).

The reproducibility dispersion is $S_{y}^{2}=0.034$.

Then, F=2.032.

$F_{\left(0.05;\;f_{W};\;f_{u}\right)}$ is the tabular value of Fisher’s test at the 5% level of significance (*F_(t)_* = 2.77) [39].

The calculated value of Fisher’s test is smaller than the table value; therefore, condition (11) is fulfilled. Therefore, the equation adequately describes the composition formulation.

After carrying out the transformation according to Equation (13), the following regression equation with the natural value of the variable parameters is obtained:$$W=10.74-0.05q_{1}+0.112q_{2}-0.0028q_{2}^{2}\;$$

The geometric interpretation of the response surface is shown in Figure 4, Figure 5 and Figure 6.

The obtained results indicate that both factors of the regression equation are significant. It should be noted that the initial parameters of the composite are affected by the linear dependence of the first factor and the quadratic dependence of the second. In the process of analysis, it is established that the impact resilience indicators take their maximum values with the content of fillers: iron slag—60–70 wt% and Waltrop—10–20 wt% (*W* = 8.5–8.7 kJ/m^2^). With a further increase in the content of particles, a deterioration of the impact resilience indicators is observed. This may be a consequence of the aggregation of fillers in the polymer matrix, which negatively affects the physical and mechanical properties of the material. Therefore, it is advisable to introduce the two-component polydispersed filler with the above-mentioned content into the modified epoxy matrix to improve the operational characteristics during the repair of the elements of the means of transport.

So, by the method of mathematically planning the experiment, the critical content of the two-component polydispersed filler is established: iron slag (*d* = 60–63 μm)—60–70 wt% and Waltrop (*d* = 8–10 μm)—10–20 wt% per 100 wt% of epoxy oligomer ED-20. The introduction of the two-component polydispersed filler into the epoxy binder makes it possible to significantly increase the indicators of the elastic modulus during the bending of protective coatings to *E* = 5.5–5.8 GPa and the impact resilience to *W* = 8.5–8.7 kJ/m^2^. The obtained results make it possible to create materials with improved overall indicators of physical and mechanical properties. It is advisable to use the obtained materials in the form of protective coatings to improve operational characteristics and to repair parts of the transport equipment.

Therefore, introducing a two-component polydispersed filler into the epoxy binder allows us to significantly increase the indicators of the modulus of elasticity during the bending of protective coatings from *E* = 4.0–4.4 GPa (composites containing one filler) to *E* = 5.5–5.8 GPa and the impact viscosity from *W* = 7.7–8.2 kJ/m^2^ (composites containing one filler) to *W* = 8.5–8.7 kJ/m^2^.

### 3.2. The Results of the Study of the Corrosion Resistance of Polymer CM

The development of the industry creates the conditions for the creation of new materials with increased operational characteristics in the complex. At the same time, the anticorrosion properties of protective composite coatings (CC), as well as the wear resistance of finished products, are of great importance. In this context, the use of epoxy-based polymer CCs is effective. Such materials are characterized by improved thermophysical, physico-mechanical, and adhesive properties and minor residual stresses [40,41,42,43,44]. The predicted control of the technological methods of introducing ingredients of different activities, dispersions, and natures under critical content at the initial stage of the formation of compositions will allow the formation of protective coatings with improved anticorrosion properties [45,46].

The analysis of existing publications involving the direction of the research into the properties of epoxy CMs indicates the need to introduce plasticizers, modifiers, and fillers into the binder to obtain epoxy polymers with increased operational characteristics. It is proven [47,48,49,50] that the introduction of modifiers and dispersed particles into the binder, even in small quantities, ensures a change in the speed of the physicochemical processes during the structuring of CM. Moreover, not only the chemical and physical natures but also the content and dispersion of the particles are important in regulating the crosslinking processes of epoxy CMs. As stated in works [36,51], it is advisable to introduce finely dispersed particles (8–30 μm) into the binder to improve the adhesive properties, and it is necessary to use a dispersed filler (63–120 μm) to improve the cohesive strength of composites. Taking this into account, it is considered expedient to conduct relevant studies to establish the influence of the nature of nano- and microdispersed fillers on the anticorrosion properties of epoxy CMs.

Developed protective coatings based on a modified epoxy binder with a three-component dispersed filler, the contents of which are determined as a result of previous research, are studied for the anticorrosion protection of technological equipment that is operated in aggressive environments (Table 4).

Five compositions of anticorrosion coatings are tested:Matrix (control sample) (the matrix is formed using the following ratio of components: epoxy oligomer ED-20 to hardener PEPA: 100:10);CM 1 (the composite is formed according to the following ratio of components: matrix to nanopowder, which is a mixture of nanodispersed compounds (MNDC), (30–90 nm) to iron scale (IS) (60–63 μm) to Waltrop (WT) (8–12 μm): 100:0.25:70:10);CM 2 (the composite is formed according to the following ratio of components: binder to MNDC to IS to WT: 100:0.25:60:20);CM 3 (the composite is formed according to the following ratio of components: binder to MNDC to IS to Agocel S-2000 (AC) (8–12 μm): 100:0.25:60:30);CM 4 (the composite is formed according to the following ratio of components: binder to MNDC to IS to AC: 100:0.25:70:20).

It is experimentally established that water sorption by epoxy composites significantly depends on the nature and cohesive strength of the coatings (Figure 7). It is found that composites filled with iron slag and Waltrop particles are the most waterproof, and composites containing iron slag particles and Agocel S-2000 are characterized by significant water sorption. The obtained results are in good agreement with the results of testing the cohesive properties of the investigated coatings—the highest water resistance is observed in coatings with significant cohesive strength and insignificant indicators of residual stresses. It is established that, at the initial stage of the study (up to 150 days), intensive swelling of all protective coatings is observed without exception. It is assumed that the aggressive environment penetrates not only into the surface layer of coatings but also into their volume. As a result, at this stage of research, the relative mass of protective coatings is *χ* = 0.5–1.2%. It is proven (Figure 7) that samples from the epoxy matrix have the maximum permeability (*χ* = 1.8%) during the determined interval of the study (*t* = 150 days). In our opinion, the obtained results are due to intensive penetration of molecules of an aggressive environment not only into the volume of the polymer but also into the metal base. At the same time, the “polymer-substrate” connections are replaced by “environment-substrate”. It is known [10] that adhesive bonds in such a tense state eventually break down, forming microcracks, since, according to the Rebinder wedging effect, an aggressive environment, penetrating the crack, causes its further development. Compared to the studied protective coatings, the original epoxy matrix is characterized by the lowest indicators of adhesive and cohesive strength, as well as significant residual stresses. As a result of the penetration of an aggressive environment into the volume of the polymer at the initial stage, its swelling occurs, later, according to the Rebinder effect; microcracks are formed and, then, a network of macrocracks. This, in turn, leads to the intensification of the swelling process, which is observed mainly during time *t* = 50–100 days of the study.

Further analysis of the research results allows us to state that during time *t* = 150–300 days of the study, the relative mass of the samples (CC 1–CC 4) practically does not change (the values are within the experimental error). However, it should be noted that during this research period, the relative mass of samples from the original epoxy matrix increases from *χ* = 1.8 to *χ* = 1.9%. This indicates the continuation of the process of the penetration of an aggressive environment into the polymer, which leads to further destruction of chemical bonds and, as a result, the peeling of the coating from the base.

A comparative analysis of the behavior of the studied composites (CC 1–CC 4) under the influence of an aggressive environment in natural conditions allows us to state that the rate of water diffusion into the polymer composite is determined mainly by diffusion at the “polymer-filler” separation boundary. It is previously proven that CC 1 and CC 3 materials have the best indicators of adhesive and cohesive strength among all the studied materials. It is established (Figure 7) that the protective coating filled with particles of a mixture of nanodispersed compounds (30–90 nm) (*q* = 0.25 wt%), iron scale (60–63 μm) (*q* = 70 wt%), and Waltrop (8–12 μm) (*q* = 10 wt%) has the lowest permeability indicators. The permeability in natural conditions of such a coating during time *t* = 250–300 days of the study is *χ* = 0.5%, which is 3.6 times less than the similar indicators of the epoxy matrix. This is due to the increased cohesive and adhesive strength of the developed coating to the metal base, which is decisive in the formation of adhesives with improved anticorrosion properties.

It is interesting from a scientific and practical point of view to conduct additional studies of the corrosion resistance of the developed materials in various aggressive environments. Therefore, in order to confirm the above results, the permeability of the composites is tested in the following environments: gasoline, acetone, NaOH (50%), I-20A lubricant, and H_2_SO_4_ (10%) (Figure 8a–f).

The results of experimental studies show the high chemical stability of materials based on an epoxy binder in various aggressive environments. It is established that the most aggressive environment for the polymer matrix is a solution of sulfuric acid (Table 14). This is due to the fact that the significant destruction of the matrix in the acid solution occurs because of the sorption of the components of the aggressive environment by the polymer, while the structure of the matrix changes and the physical and chemical bonds are destroyed.

It is established (Table 14) that the introduction of dispersive fillers into the epoxy oligomer helps to increase the chemical resistance of the materials. This is due to a decrease in the relative content of the polymer in the volume of the composite and an increase in the path of penetration of the molecules of corrosive agents during diffusion to the metal base because of the barrier effect created by the dispersed particles of the filler. The obtained results of the materials research in aggressive environments correlate with similar results of tests of the same composites in natural conditions. It is established (Table 14) that the CC 1 composite has the best indicators of chemical resistance among all the studied materials. It is proven that the formation of an epoxy-based coating with particles of iron slag and Waltrop at the optimal content (respectively, *q* = 70 wt% and *q* = wt% for *q* = 100 wt% of ED-20 resin) increases relative to the epoxy matrix indicators of chemical resistance: 2.9 times in gasoline; 3.3 times in acetone; 2.9 times in NaOH (50%); 2.9 times in I-20A lubricant; 2.0 times in H_2_SO_4_ (10%). First of all, this is due to the cohesive interaction at the “polymer-filler” phase-separation boundary, which significantly affects the protective properties of CC. Low adhesion of dispersed particles to the polymer in the volume of the coating reduces the protective properties of the adhesive because of the development of a network of cracks, which, in most cases, is associated with the dislodging effect of an aggressive environment.

Composites filled with iron scale and, additionally, Agocel S-2000 particles have low chemical resistance, especially in I-20A lubricant and in NaOH and H_2_SO_4_ solutions. This is due to the adsorption of water and the molecules of active reagents on the surface of hydrophilic oxides in the form of hydroxyl groups held by hydrogen bonds.

Therefore, modified epoxy composite protective coatings with improved anticorrosion properties are developed in the work.

It is established that the protective coating filled with particles of a mixture of nanodispersed compounds (30–90 nm) (*q* = 0.25 pts.wt.), iron scale (60–63 μm) (*q* = 70 pts.wt.), and Waltrop (8–12 μm) (*q* = 10 pts.wt.) has the lowest permeability indicators. The permeability in natural conditions of such a coating during time t = 250–300 days of the study is *χ* = 0.5%, which is 3.6 times less than the similar indicators of the epoxy matrix. This is due to the increased cohesive and adhesive strength of the developed coating to the metal base, which is decisive in the formation of adhesives with improved anticorrosion properties.It is proven that the formation of an epoxy-based coating with particles of iron slag and Waltrop at the optimal content (respectively, *q* = 70 pts.wt. and *q* = 10 pts.wt. for *q* = 100 pts.wt. of ED-20 resin) increases relative to the initial chemical resistance indicators of the epoxy matrix: 2.9 times in gasoline; 3.3 times in acetone; 2.9 times in NaOH (50%); 2.9 times in I-20A lubricant; 2.0 times in H_2_SO_4_ (10%). This is due to the cohesive interaction at the “polymer—filler” phase-separation boundary, which significantly affects the protective properties of the adhesive.

### 3.3. The Results of the Study of Hydroabrasive Wear Resistance of Polymer CMs

At the next stage, the wear resistance of the developed materials under the action of hydroabrasion is investigated (Figure 9a–f).

The analysis of the coefficient of wear resistance of the studied CMs at the angle of attack of the hydroabrasive mixture *å* = 45° allows us to state that the wear resistance is relatively high in all samples, without exception. It is experimentally established that the initial epoxy matrix has the lowest coefficient of wear resistance among all investigated composites (Figure 10). It is shown that at the angle of attack of the hydroabrasive *å* = 45°, the wear resistance coefficient of the matrix is *K* = 1.32. The authors [11,24,52,53,54] proved that the intensity of wear depends on various destruction processes: knocking out the coating surface with an abrasive mixture and its deformation. At the same time, the dominant influence of each of the processes depends on the cohesive strength of the CM. The method of optical microscopy proved that the matrix is characterized by two types of destruction of the surface layer of the material: macrocutting and plastic deformation with subsequent removal of the material.

Additionally, it is established (Figure 10) that the CC 3 composite is characterized by the highest indicators of resistance to hydroabrasive operation, for which the following indicator of wear resistance is observed: *K* = 1.75. It is confirmed by the method of optical microscopy that branched relief bands are observed on the surface of such a composite, which arose as a result of the action of abrasive water particles (Figure 11a). At the same time, compared to the matrix, they are not so deep, and their length is much shorter. The improvement in the wear resistance of epoxy composites can be explained by their improved cohesive properties compared to the epoxy matrix, which implies an increase in the wear resistance coefficient.

Composites filled with particles of iron slag and Waltrop (CC 1, CC 2) are characterized by slightly worse indicators of wear resistance. It is proved (Figure 10) that at the angle of attack of hydroabrasive *å* = 45°, their wear resistance increases in relation to the matrix from *K* = 1.32 to *K* = 1.54–1.68. At the same time, the course of the process of knocking out the surface of the coating with an abrasive mixture is observed, which is typical for the wear of fragile heterogeneous polymer composites. The method of optical microscopy confirmed (Figure 11b,c) that multiple deformations of CM as a result of micro-impacts by abrasive particles lead to the formation of microcracks on the surface, the plane of which is perpendicular to the direction of movement of the hydroabrasive mixture. In this case, grooves appear on the contact surface that have an orientation in the direction of the sliding speed vector of the hydroabrasive flow [55,56,57]. Mostly, such a mechanism makes an insignificant contribution to the activation process, compared to the mechanism of plastic deformation, which is dominant for epoxy matrix. When implementing such a trigger mechanism, individual abrasive particles under the influence of a significant nominal pressure of the hydroabrasive flow are wedged into the surface layer of the CM, plastically deforming the epoxy matrix. As a result, an area of compressive deformation (in front of the contact area) and tensile deformation (behind the contact area) is formed, which leads to the formation of different sizes of microcracks. Under the influence of the tangential force of the hydroabrasive flow, the abrasive particle, together with part of the polymer, is removed from the surface of the material. Multiple deformations of the polymer composites containing dispersed particles as a result of subsequent activation lead to fatigue and local removal of the epoxy composite. This leads to the formation of new microcracks. Over time, the destruction is localized in those areas of the CM that have the highest density of microcracks. At the same time, it is possible to expect the appearance of a macrocrack oriented perpendicular to the direction of the speed of movement of the hydroabrasive mixture. Under the influence of repeated maximum deformations, there is an increase in macro-destroyed areas with removed polymer, which leads to the formation of a wavy relief called the “Shallomach pattern”. The waves observed by optical microscopy on the surface of CM samples (Figure 11b,c) are located perpendicular to the direction of movement of the abrasive particles, and the pattern of the triggering surface is preserved even after repeated studies.

The standard is Ct 3 steel.

So, according to the test results, it is established that the protective coating filled with particles of a mixture of nanodispersed compounds (30–90 nm) (*q* = 0.25 wt%), iron scale (60–63 μm) (*q* = 60 wt%), and Agocel S-2000 (8–12 μm) (*q* = 30 wt%) have the highest indicators of wear resistance. The coefficient of wear resistance under the action of hydroabrasion of such a coating is *K* = 1.75, which is 1.3 times higher than the similar indicators of the epoxy matrix. It is shown that the mechanism of the wear of materials is determined by physical and mechanical processes on the surface of composites, the determining factors of which are the processes of knocking out microparticles of the surface of the coating with an abrasive mixture and the formation of a wavy relief.

In addition, SEM images of the structure of the studied CMs are provided in the work for a detailed analysis of the structure of the formed materials. It is considered appropriate to present the results of the study of the structure of the CM, which are shown in Figure 11, i.e., CC 3, CC 1, and CC 2.

The analysis of the SEM image of the fracture surface of CC 3 deserves special attention (Figure 12a). Such coatings are characterized by a finer and more uniform topology of the fracture surface, which makes it possible to assume the presence of stoppers for the propagation of microcracks, through which particles of nano- and microfillers can protrude in a uniformly structured system of “polymer—dispersed particles”. Due to the organization of the nanolevel system of microcrack stoppers, the energy expended on the destruction of materials increases (Figure 10 and Figure 11). At the same time, indicators of the coefficient of hydroabrasive wear resistance of CM are increasing. It can be argued that the propagation of main cracks, which originate at the point of impact of the hydroabrasive particles, is inhibited, which further confirms the above statements.

Analysis of the SEM image of the fracture surface of the CC 1 material (Figure 12b) similarly revealed a fine but heterogeneous topology of the fracture surface.

Additionally, the fracture surface of the CC 2 material is analyzed (Figure 12c). Analysis of the SEM image revealed the inhomogeneity of the veneers with the chaotic direction of crack propagation, which indicates the instability of the operational characteristics of such materials.

### 3.4. Protective Epoxy Composite Coatings with Increased Operational Characteristics for the Repair of Parts of Transport Equipment

On the basis of the conducted research, materials and modes of formation of epoxy compositions for protective coatings with increased operational characteristics are developed. The developed polymer composite coatings (PCC) that meet high operational requirements include PCC-1 and PCC-2.

Coating 1 (PCC-1). The main purpose of the coating is to improve the anticorrosion properties of technological equipment. PCC-1 is a material based on epoxy matrix and nano- and microdispersed filler. The developed material has high performance characteristics and anticorrosion properties, and its service life is 3–5 years. The low cost of the ingredients of the polymer composition, compared to known materials, is ensured by the increase in quality and the increase in the service life and inter-repair periods of work.

The technological process of forming PCC-1 consists of the following operations: surface preparation, preparation of compositions, application of adhesive and surface layers, and polymerization of the material.

The quality of the preparation of the protective surface largely determines the reliability and durability of PCC-1. Surface preparation consists of degreasing and the removal of various impurities, scale, and rust by sandblasting.

Preparation of compositions consists of the dosing of components and the preparation of fillers (purification of dispersed particles from impurities by ultrasonic treatment). Powders of homogeneous fractions are dried in an oven at a temperature of 353–363 K for 2 h.

Fillers are added to the epoxy resin in appropriate proportions and mixed. After the hydrodynamic mixing of the components, the hardener is introduced immediately before applying the composition to the steel surface.

Coating 1 (PCC-1) consists of the following components, pts.wt.:-Epoxy dian oligomer ED-20—100;-Polyethylene polyamine (PEPA) hardener—10.

Its filler consists of the following:
-Nanodispersed filler in the form of a mixture of nanodispersed compounds (Si_3_N_4_, I_2_O_3_, and Al_2_O_3_) (MNDC), (30–90 nm)—0.25–0.50;-Waltrop (WT), (8–12 μm)—10–20;-Iron scale (IS), (60–63 μm)—60–70.

The coating is applied in traditional ways. The most productive and technological method is the method of pneumatic spraying, which allows the application of uniform layers of material on the surface of a complex configuration.

Coating 2 (PCC-2). The main purpose is to increase the resistance to the hydroabrasive operation of technological equipment. The developed material has high physical, mechanical, and thermophysical properties and wear resistance, and its service life is 4–6 years.

The technological process of forming PCC-2 consists of the operations given above when describing the technology of forming the cover of PCC-1. After hydrodynamically mixing the components, the hardener is introduced immediately before applying the composition to the steel surface.

Coating 2 (PCC-2) consists of the following components, pts.wt.:-Epoxy dian oligomer ED-20—100;-Polyethylene polyamine (PEPA) hardener—10.

Its filler consists of the following:-Nanodispersed filler in the form of a mixture of nanodispersed compounds (Si_3_N_4_, I_2_O_3_, and Al_2_O_3_) (MNDC), (30–90 nm)—0.10–0.25;-Agocel S-2000 (AC), (8–12 μm)—20–30;-Iron scale (IS), (60–63 μm)—60–70.

The coating is applied by the method of pneumatic spraying, which allows the forming of uniform layers of material on the surface of a complex configuration.

The results of comparative tests of physico-mechanical, thermo-physical, and anticorrosion properties and resistance to the hydroabrasive action of developed materials and protective coatings based on them testify to the high operational characteristics and the feasibility of using new composites for the repair of equipment and sea and river transport (Table 15).

## 4. Conclusions

This paper developed a new approach to solving an important scientific and technical problem, which consists of the development of modified protective coatings filled with nano- and microdispersed particles with increased operational characteristics for the restoration of transport. The solution to the scientific problem consists of scientifically based management of the processes of interphase interaction as a result of the predicted introduction of additives into the binder at an optimal content, as well as in establishing the regularities of the interrelationship of anticorrosion properties and wear resistance with the structure of materials. As a result of this work, the following main conclusions are obtained.

A mathematical model is developed for optimizing the content of components in the formation of protective anticorrosion and wear-resistant coatings for means of transport as a result of the complex effect of a mixture of nanodispersed compounds, iron scale, and Waltrop. Using the method of mathematically planning the experiment using the STATGRAPHICS^®^ Centurion XVI application package, the content of additives of different physical and chemical natures in the epoxy binder is optimized to obtain protective coatings with improved operational characteristics. It is proved that the introduction of iron scale (*d* = 60–63 μm)—60–70 pts.wt. and Waltrop (*d* = 8–12 μm)—10–20 pts.wt. per 100 pts.wt. of oligomer ED- 20 and 10 pts.wt. of the PEPA hardener ensures the formation of a material with a bending modulus of elasticity of *E* = 5.5–5.8 GPa and an impact resilience of *W* = 8.5–8.7 kJ/m^2^. That is, optimization of the content of fillers of different physical and chemical nature in the epoxy binder provides a 1.3 increase in physical and mechanical properties (compared to an unfilled matrix), which makes it possible to use such materials in the formation of functional protective coatings.

Modified epoxy composite protective coatings with improved anticorrosion properties and wear resistance under hydroabrasive conditions are developed. It is established that the protective coating filled with particles of a mixture of nanodispersed compounds (30–90 nm) (*q* = 0.25 wt%), iron scale (60–63 μm) (*q* = 70 wt%), and Waltrop (8–12 μm) (*q* = 10 wt%) has the lowest permeability indicators. The permeability in natural conditions of such a coating during time *t* = 300 days of the study is *χ* = 0.5%, which is 3.6 times less than the similar indicators of the epoxy matrix. A comparative analysis of the behavior of the studied composites under the influence of an aggressive environment in natural conditions allows us to state that the rate of diffusion of water into the polymer composite is determined mainly by diffusion at the border of the “polymer—filler” separation. It is previously proven that the material of this composition has the best indicators of adhesive and cohesive strength among all the studied materials. This is due to the increased cohesive and adhesive strength of the developed coating to the metal base, which is decisive in the formation of adhesives with improved anticorrosion properties.

It is proven that the formation of an epoxy-based coating with particles of iron slag and Waltrop at the optimal content (respectively, *q* = 70 wt% and *q* = 10 wt% for *q* = 100 wt% of ED-20 resin) increases relative to the initial indicators of chemical resistance of the epoxy matrix: 2.9 times in gasoline; 3.3 times in acetone; 2.9 times in NaOH (50%); 2.9 times in I-20A lubricant; 2.0 times in H_2_SO_4_ (10%). First of all, this is due to the cohesive interaction at the boundary of the “polymer—filler” phase separation, which significantly affects the protective properties of the coating. The high adhesion of dispersed particles to the polymer in the volume of the coating increases the protective properties of the adhesive because of a decrease in the rate of development of the crack network. The latter, in most cases, is associated with the disintegrating effect of an aggressive environment.

It is substantiated that the protective coating filled with particles of a mixture of nanodispersed compounds (30–90 nm) (*q* = 0.25 wt%), iron scale (60–63 microns) (*q* = 60 wt%), and Agocel S-2000 (8–12 μm) (*q* = 30 wt%) is characterized by the highest indicators of wear resistance. The coefficient of wear resistance under the action of hydroabrasion of such a coating is *K* = 1.75, which is 1.3 times higher than the similar indicators of the original epoxy matrix. It is shown that the mechanism of material wear is caused by physical and mechanical processes on the surface of composites, the determining factors of which are the processes of microcutting and plastic deformation of the surface layer. Using optical and electron microscopy, it is confirmed that branched relief bands are observed on the surface of such a composite, which arose as a result of the action of hydroabrasive particles. At the same time, compared to other studied materials, they are not so deep, and their length is significantly shorter. The improvement in the wear resistance of epoxy composites can be explained by their improved cohesive properties compared to the epoxy matrix, which implies an increase in the wear resistance coefficient.

## 5. Prospects for Future Research

Based on the above, it is necessary to state the feasibility of developing new polymer composites, which would be characterized by increased operational characteristics in the complex. To improve the properties of polymers, including those based on epoxy, it is advisable to introduce modifiers into the oligomer, since such additives with a small content (1–3%) allow a significant increase in the functional properties of composites. At the same time, it is important to choose a modifier that contains hydroxyl, carbonyl, nitrile, and other groups that are active for physical and chemical interaction with the epoxy oligomer. It is important to establish the critical content of the additive in the polymer, since its excessive amount leads to an increase in the sol fraction in the materials, which leads to the deterioration of their cohesive strength.

Taking into account that the developed materials can be effectively used to protect equipment that is operated under variable loads and at elevated temperatures, it is considered appropriate to conduct a study on determining the effect of the nature and content of the modifier on the properties of the epoxy matrix. In addition, it is necessary to state the feasibility of developing new polymer composite materials, which would be characterized by improved adhesive, physical-mechanical, and thermophysical properties. This is achieved by introducing a modifier and dispersed fillers of various natures. This approach will allow the obtaining of protective coatings, which are appropriate and necessary to use when restoring parts of machines and mechanisms of means of transport.

## Figures and Tables

**Figure 1 polymers-15-03449-f001:**
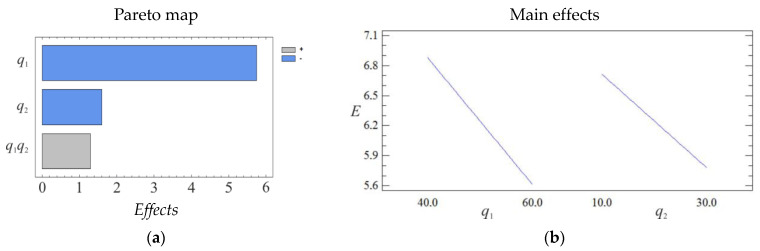
Pareto map (**a**) and main effects (**b**) of *E*.

**Figure 2 polymers-15-03449-f002:**
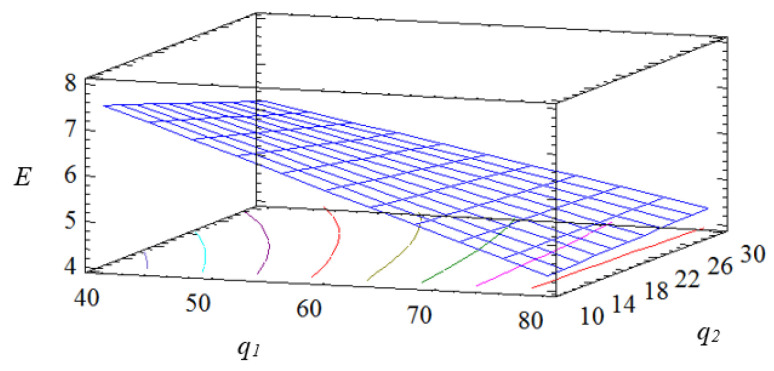
Calculated response surface *E* = *f* (*q_1_*, *q_2_*).

**Figure 3 polymers-15-03449-f003:**
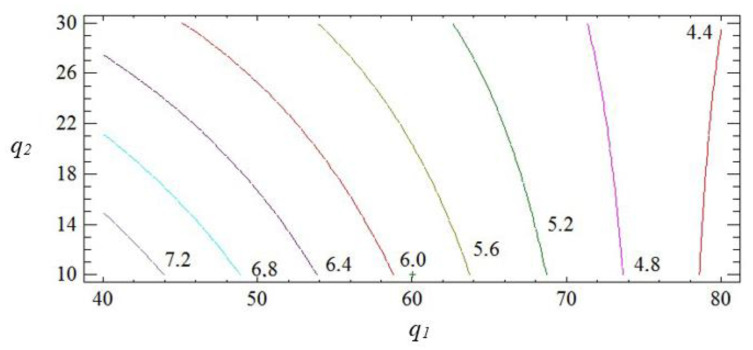
Contours of the calculated response surface.

**Figure 4 polymers-15-03449-f004:**
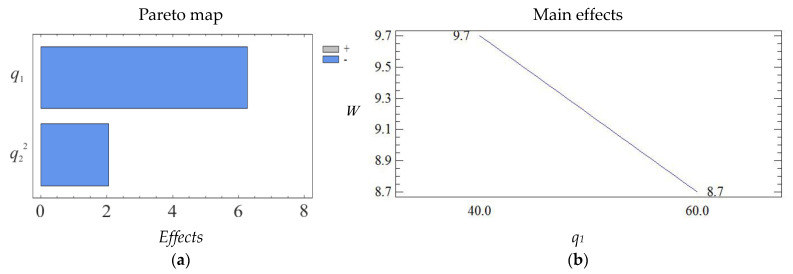
Pareto map (**a**) and main effects (**b**) of *W*.

**Figure 5 polymers-15-03449-f005:**
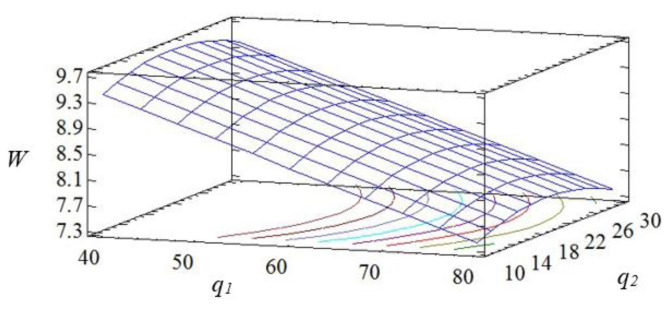
Calculated response surface *W = f (q*_1_*, q*_2_*)*.

**Figure 6 polymers-15-03449-f006:**
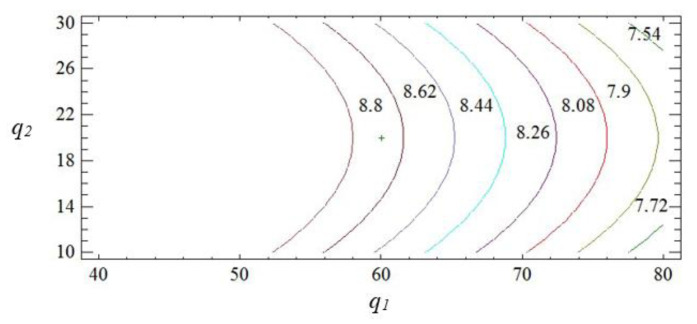
Contours of the calculated reference surface.

**Figure 7 polymers-15-03449-f007:**
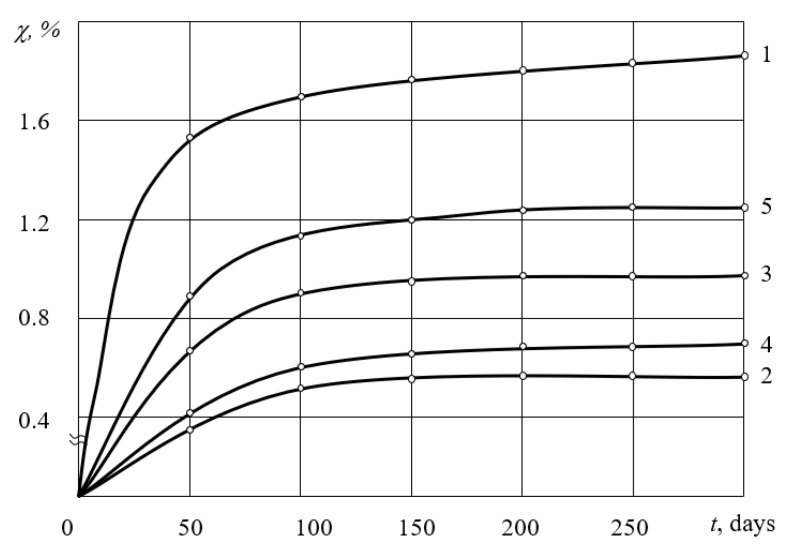
Dependence of water sorption in natural conditions by protective coatings on duration of exposure: 1—matrix (control sample); 2—CC 1; 3—CC 2; 4—CC 3; 5—CC 4.

**Figure 8 polymers-15-03449-f008:**
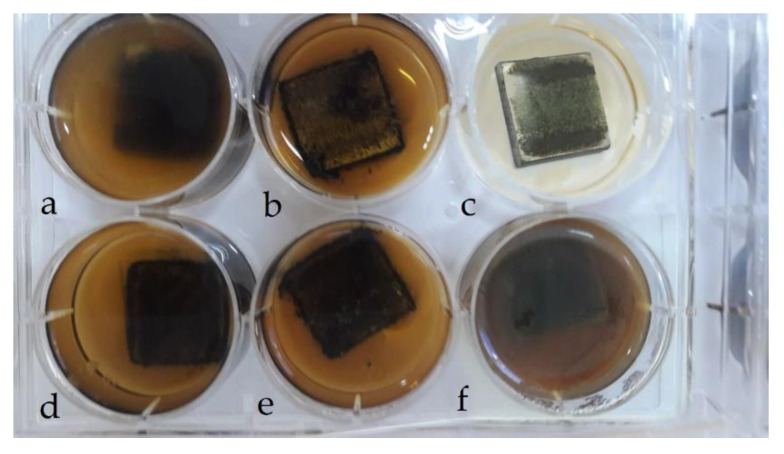
Determination of permeability index in aggressive environments: (**a**)—sulfuric acid (CAS No. 7664-93-9); (**b**)—solutions of sodium (CAS No. 1310-73-2); (**c**)—technical water (CAS No. 7732-18-5); (**d**)—gasoline (CAS No. 64742-82-1); (**e**)—acetone (CAS No. 67-64-1); (**f**)—I-20A lubricant (CAS No. 64742-62-7).

**Figure 9 polymers-15-03449-f009:**
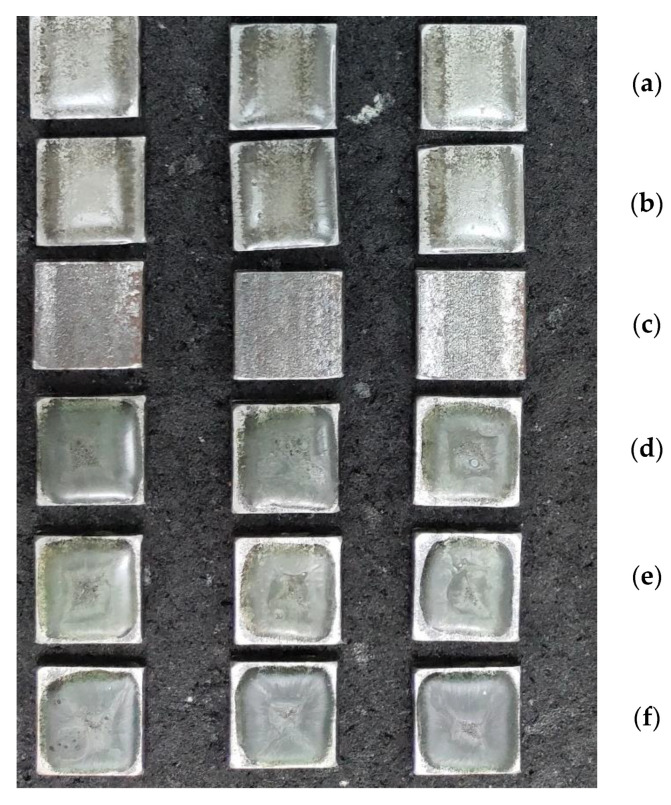
General appearance of the formed samples used in the study of hydroabrasive wear resistance: (**a**)—epoxy matrix; (**b**)—CC 1; (**c**)—Ct 3 steel; (**d**)—CC 2; (**e**)—CC 3; (**f**)—CC 4.

**Figure 10 polymers-15-03449-f010:**
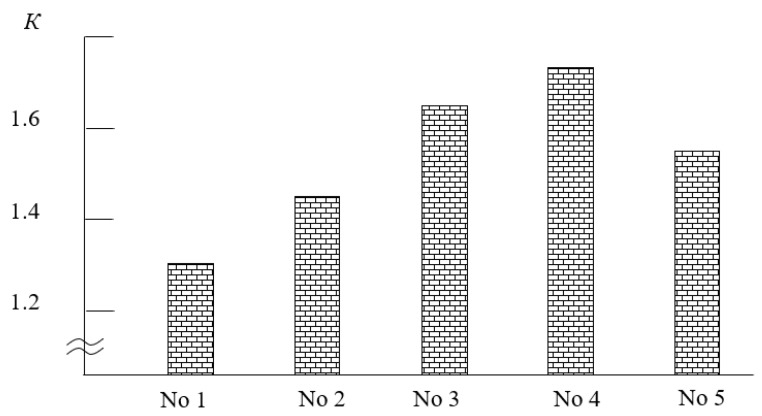
Dependence of the coefficient of wear resistance (K) at the angle of attack of the hydroabrasive *å* = 45° on the content and nature of the ingredients in the composites: No. 1—matrix (control sample); No. 2—CC 1; No. 3—CC 2; No. 4—CC 3; No. 5—CC 4.

**Figure 11 polymers-15-03449-f011:**
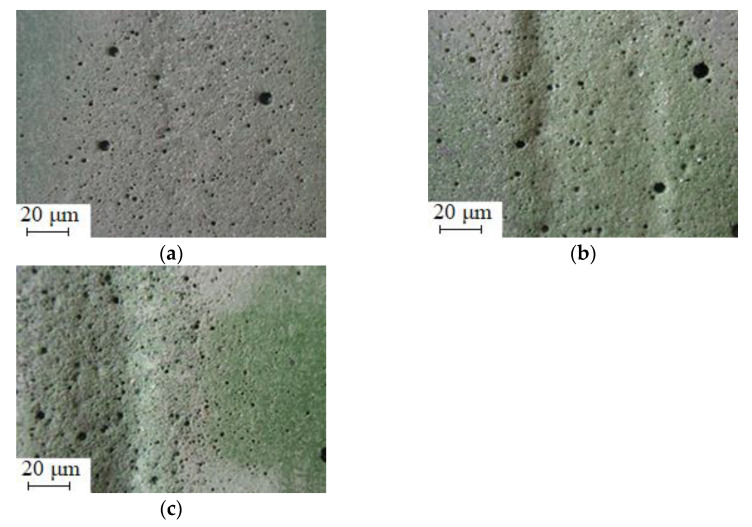
Appearance of the surface after hydroabrasive destruction of CM containing (**a**) CC 3, (**b**) CC 1, and (**c**) CC 2.

**Figure 12 polymers-15-03449-f012:**
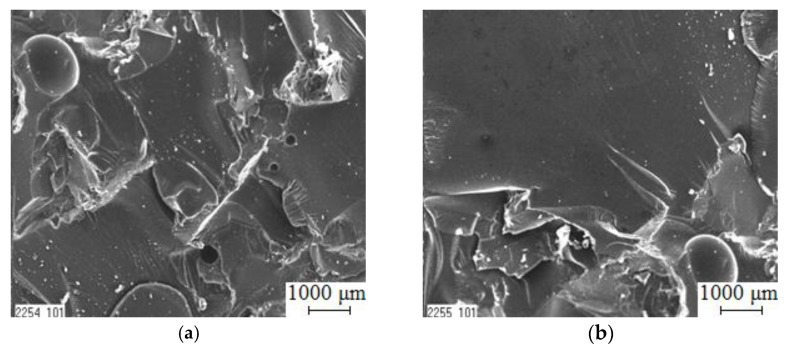
Electron micrographs of the structure of epoxy composites: (**a**) CC 3; (**b**) CC 1; (**c**) CC 2.

**Table 1 polymers-15-03449-t001:** Characteristics of MNDC.

Characteristics	Si_3_N_4_	I_2_O_3_	Al_2_O_3_
Specific surface area, S, m^2^/g	44	14	44
Particle size, determined by the method of thermal adsorption, *d*, nm	41	89	41
Particle size determined by electron microscopy, *d*, nm	39	35	76

**Table 2 polymers-15-03449-t002:** Characteristics of fillers.

Characteristics	Agocel S-2000	Waltrop
Shape	Round powder particles	Round powder particles
Color	White	Yellow
Smell	Absent	Absent
Destruction temperature, °C	140	220
Flowability density, *ρ*, kg/m^3^	660	520
Solubility in water at 20 °C, g/L	5	12
pH value at 20 °C	5–7	6–8
Dynamic viscosity at 20 °C, *η*, mPa·s	1500	1850

**Table 3 polymers-15-03449-t003:** Levels of variables in conventional and natural scales.

Components	Factor	Medium Level, *q*, pts.wt.	Step of Variation, Δ*q*, pts.wt.	Values of Variable Levels (pts.wt.) Corresponding to Conventional Units		
				−1	0	+1
The main filler— iron slag	*x* _1_	70	10	60	70	80
Additional filler— Waltrop	*x* _2_	20	10	10	20	30

**Table 4 polymers-15-03449-t004:** Experiment planning scheme.

No. *(u)*	*x* _0_	*x* _1_	*x* _2_	$x_{3}=x_{1}^{2}-d$	$x_{4}=x_{2}^{2}-d^{\;}$	$x_{1}x_{2}\;$
1	1	−1	−1	0.33	0.33	+1
2	1	+1	−1	0.33	0.33	−1
3	1	−1	+1	0.33	0.33	−1
4	1	+1	+1	0.33	0.33	+1
5	1	0	0	−0.67	−0.67	0
6	1	+1	0	0.33	−0.67	0
7	1	−1	0	0.33	−0.67	0
8	1	0	+1	−0.67	0.33	0
9	1	0	−1	−0.67	0.33	0
$\sum_{u-1}^{N}x_{iu}^{2}$	9	6	6	2	2	4

**Table 5 polymers-15-03449-t005:** The extended planning matrix of the full factorial experiment.

No.	Content of Components, *q*, pts.wt.		Elastic Modulus, *E*, GPa	Impact Resilience, W*,* kJ/m^2^
	*x* _1_	*x* _2_	*y* _1_	*y* _2_
1	60	10	5.8	8.5
2	80	10	4.0	7.3
3	60	30	5.5	8.1
4	80	30	4.4	7.4
5	70	20	5.0	8.3
6	80	20	4.6	7.6
7	60	20	5.5	8.7
8	70	30	4.6	8.2
9	70	10	5.4	8.0

**Table 6 polymers-15-03449-t006:** Coefficients of the regression equation.

*b* _0_	*b* _1_	*b* _2_	*b* _11_	*b* _22_	*b* _12_
5.06	−0.63	−0.12	−0.03	−0.08	0.18

**Table 7 polymers-15-03449-t007:** Values of adequacy variance ($S_{ui}^{2}$ ) and reproduced variances ($\sigma^{2}{\left\{y\right\}}_{i}$ ).

No.	Dispersion of Adequacy		Dispersion of Reproduction	
	Conventional Symbol	Value	Conventional Symbol	Value
1	$S_{u1}^{2}$	0.01	$\sigma^{2}{\left\{y\right\}}_{1}$	0.02
2	$S_{u2}^{2}$	0.01	$\sigma^{2}{\left\{y\right\}}_{2}$	0.02
3	$S_{u3}^{2}$	0.01	$\sigma^{2}{\left\{y\right\}}_{3}$	0.02
4	$S_{u4}^{2}$	0.03	$\sigma^{2}{\left\{y\right\}}_{4}$	0.06
5	$S_{u5}^{2}$	0.01	$\sigma^{2}{\left\{y\right\}}_{5}$	0.02
6	$S_{u6}^{2}$	0.03	$\sigma^{2}{\left\{y\right\}}_{6}$	0.06
7	$S_{u7}^{2}$	0.04	$\sigma^{2}{\left\{y\right\}}_{7}$	0.06
8	$S_{u8}^{2}$	0.01	$\sigma^{2}{\left\{y\right\}}_{8}$	0.02
9	$S_{u9}^{2}$	0.01	$\sigma^{2}{\left\{y\right\}}_{9}$	0.02

**Table 8 polymers-15-03449-t008:** Experimental results of the study of the elasticity modulus of CM.

No. of Research	Elasticity Modulus, *E*, GPa			Average Value, *E*, GPa
	1	2	3	
1	5.7	5.8	5.9	5.8
2	3.9	4.1	4.0	4.0
3	5.5	5.6	5.4	5.5
4	4.3	4.3	4.6	4.4
5	5.1	4.9	5.0	5.0
6	4.5	4.8	4.5	4.6
7	5.3	5.6	5.6	5.5
8	4.5	4.7	4.6	4.6
9	5.3	5.5	5.4	5.4

**Table 9 polymers-15-03449-t009:** Dispersions of regression coefficients ($S_{b_{}}^{2}$ ) and calculated values of Student’s test (*t_c_*).

No.	Dispersions of Regression Coefficients		Calculated Values of Student’s Test	
	Conventional Symbol	Value	Conventional Symbol	Value
1	$S_{b_{0}}^{2}$	0.002	*t* _0*p*_	116.63
2	$S_{b_{1}}^{2}$	0.003	*t* _1*p*_	12.02
3	$S_{b_{2}}^{2}$	0.003	*t* _2*p*_	2.21
4	$S_{b_{11}}^{2}$	0.008	*t* _11*p*_	0.37
5	$S_{b_{22}}^{2}$	0.008	*t* _22*p*_	0.91
6	$S_{b_{12}}^{2}$	0.004	*t* _12*p*_	2.70

**Table 10 polymers-15-03449-t010:** Coefficients of the regression equation for impact resilience.

*b* _0_	*b* _1_	*b* _2_	*b* _11_	*b* _22_	*b* _12_
8.36	−0.50	−0.02	−0.23	−0.28	0.13

**Table 11 polymers-15-03449-t011:** Adequacy dispersion value ($S_{u}^{2}$ ) and reproduction dispersion value (*σ^2^(y)*).

No.	Adequacy Dispersion		Reproduction Dispersion	
	Conventional Symbol	Value	Conventional Symbol	Value
1	$S_{u1}^{2}$	0.01	$\sigma^{2}{\left\{y\right\}}_{1}$	0.02
2	$S_{u2}^{2}$	0.07	$\sigma^{2}{\left\{y\right\}}_{2}$	0.14
3	$S_{u3}^{2}$	0.04	$\sigma^{2}{\left\{y\right\}}_{3}$	0.08
4	$S_{u4}^{2}$	0.07	$\sigma^{2}{\left\{y\right\}}_{4}$	0.14
5	$S_{u5}^{2}$	0.03	$\sigma^{2}{\left\{y\right\}}_{5}$	0.06
6	$S_{u6}^{2}$	0.01	$\sigma^{2}{\left\{y\right\}}_{6}$	0.02
7	$S_{u7}^{2}$	0.04	$\sigma^{2}{\left\{y\right\}}_{7}$	0.08
8	$S_{u8}^{2}$	0.03	$\sigma^{2}{\left\{y\right\}}_{8}$	0.06
9	$S_{u9}^{2}$	0.01	$\sigma^{2}{\left\{y\right\}}_{9}$	0.02

**Table 12 polymers-15-03449-t012:** Experimental results of the study of impact resilience of CM.

No. of the Experiment	Impact Resilience, *W,* kJ/m^2^			Average Value, *W,* kJ/m^2^
	1	2	3	
1	8.4	8.5	8.6	8.5
2	7.0	7.5	7.4	7.3
3	7.9	8.3	8.1	8.1
4	7.2	7.7	7.3	7.4
5	8.1	8.4	8.4	8.3
6	7.6	7.7	7.5	7.6
7	8.9	8.5	8.7	8.7
8	8.0	8.3	8.3	8.2
9	8.0	8.1	7.9	8.0

**Table 13 polymers-15-03449-t013:** Dispersions of regression coefficients ($S_{b_{}}^{2}$ ) and calculated values of Student’s test (*t_c_*).

No.	Dispersions of Regression Coefficients		Calculated Values of Student’s Test	
	Conventional Symbol	Value	Conventional Symbol	Value
1	$S_{b_{0}}^{2}$	0.004	*t* _0*p*_	132.44
2	$S_{b_{1}}^{2}$	0.006	*t* _1*p*_	6.60
3	$S_{b_{2}}^{2}$	0.006	*t* _2*p*_	0.22
4	$S_{b_{11}}^{2}$	0.017	*t* _11*p*_	1.78
5	$S_{b_{22}}^{2}$	0.017	*t* _22*p*_	2.16
6	$S_{b_{12}}^{2}$	0.009	*t* _12*p*_	1.30

**Table 14 polymers-15-03449-t014:** Relative change in mass (*χ*, %) of composites after their aging in aggressive environments (at temperature *T* = 293 ± 2 K, during *τ* = 1420 h).

Type of Coating	Aggressive Environment				
	Gasoline	Acetone	NaOH (50%)	I-20A Lubricant	H_2_SO_4_ (10%)
Matrix	5.2	5.0	5.3	6.3	8.4
CC 1	1.8	1.5	1.8	2.2	4.3
CC 2	2.4	2.8	2.6	3.2	5.5
CC 3	1.9	1.9	2.2	2.8	5.0
CC 4	2.7	2.8	2.9	3.6	6.1

**Table 15 polymers-15-03449-t015:** Comparative indicators of properties of developed composite materials and coatings based on them.

Indicator	PCC-1	PCC-2
Adhesive strength (*σ_a_*, MPa)	53	53
Destructive stresses during bending (*σ_bd_*, MPa)	54	56
Modulus of elasticity in bending (*E*, GPa)	5.8	5.3
Heat resistance (*T*, K)	364	366
Impact resilience (*W*, kJ/m^2^)	8.7	8.2
Penetration in natural conditions of the coating during time *t* = 250–300 days (*χ* %)	0.5	0.7
Wear resistance (at the angle of attack of hydroabrasive *å* = 45°), *K*	1.68	1.75

## Data Availability

Not applicable.

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
