# Peer review of "Functional Polymer Nanocomposites with Increased Anticorrosion Properties and Wear Resistance for Water Transport"

_polymers, 2023, doi:10.3390/polym15163449_

Round 1

Reviewer 1 Report (Previous Reviewer 2)

The paper has been improved a lot after revision. I think it meets the requirement of Polymers. I recommend it to publish on this journal.

none

Author Response

Thank you!

Reviewer 2 Report (New Reviewer)

The manuscript "Functional polymer nanocomposites for the restoration of water transport vehicles" by Buketov et al. showed howed a high originality of research and clear presentation of data with well-written manuscript. However, there are some minor changes required before the publication. Please check and revise followings.

1. For the Materials and Methods section, the subtitles are mixed with upper and lower cases. Please check. (2.2 Research methods, 2.3 Statistical Methods)

2. For the Materials and Methods section, there are no information regarding on the usage of instruments. For instance, is Figure 11 taken by digital photo or optical microscope? Please clarify for all other Figures.

3. For Table 15, please clear the contents of Indicator section. For example, Adhesive strength, σа, MPa to Adhesive strength (σа, MPa) 

4. For the Conclusions section, authors have concluded the paper with numbering and I am wondering that such numbering is required? 

Author Response

Response to Reviewer 2 Comments

  1. For the Materials and Methods section, the subtitles are mixed with upper and lower cases. Please check. (2.2 Research methods, 2.3 Statistical Methods)

Response 1: It is corrected in the text of the publication (lines 250).

  1. For the Materials and Methods section, there are no information regarding on the usage of instruments. For instance, is Figure 11 taken by digital photo or optical microscope? Please clarify for all other Figures.

Response 2: Fig. 8 and fig. 9 are digital photos. Fig. 11 was obtained using an XJL-17AT metallographic microscope equipped with a Levenhuk C310 NG camera (3.2 Mega Pixels) (lines 242-245).

  1. For Table 15, please clear the contents of Indicator section. For example, Adhesive strength, σа, MPa to Adhesive strength (σа, MPa) 

Response 3: It is corrected in the text of the publication (Table 15).

  1. For the Conclusions section, authors have concluded the paper with numbering and I am wondering that such numbering is required? 

Response 4: The numbering has been removed.

Reviewer 3 Report (New Reviewer)

Manuscript ID: polymers-2546668 Functional polymer nanocomposites for the restoration of water

transport vehicles” The study is novel and technically important, however intense major revisions needed for several improvements in the manuscript as:

1.    Title must be modified to better present the nature of studies?

2.    Abstract is poorly written and not reader friendly to reflect the purpose, design, and results of the composites.

3.    Please rewrite introduction to better reflect the worth of your composites and analysis methods and differences with the already reported studies.

4.    The entire manuscript must be professionally revised for the language and grammar improvements. The language and discussions throughout the manuscript is not fluent.

5.    The morphology studies using SEM, TEM, AFM, etc. must be included to enhance the worth of the materials used.

6.    Please also revise the conclusions and describe the outcomes sequentially with reasoning.

Please see above report.

Round 2

Reviewer 3 Report (New Reviewer)

The manuscript has been significantly improved along the suggested lines, so can be accepted now. 

The manuscript has been significantly improved along the suggested lines, so can be accepted now. 

This manuscript is a resubmission of an earlier submission. The following is a list of the peer review reports and author responses from that submission.

Round 1

Reviewer 1 Report

The manuscript reported preparation and characterization of epoxy composites containing nanoparticles and microparticles. The authors used statistical calculations to theoretically optimize the coating formations and then experimentally studied the corrosion resistance and wear resistance on the optimized coating formulations. The research in this work is interesting and however the quality of research is not high enough for publication in the present form. The follow queries should be addressed.

1)    Abstract should be rewritten. The abstract described the improvements of mechanical properties, and thermophysical properties of the composites. However, there are no data in the results and discussion section to support. The abstract also described the fillers changed the topological structure of the composites and a conformational set of macromolecules. However, the authors did not report any structures in this work

2)    The molecular formula should be written with subscripts.

3)    Research methods: the authors described the research methods on elastic modulus and impact resilience and however they did not report any experimental data.

4)    The first two paragraphs in the section 3.1 Mathematical planning of the experiment should be removed as they are literature review not research data.

5)    The authors used mathematical methods to calculate the elastic modulus and impact toughness of composites in order to optimize the coating formulations. However, the authors did not report any experimental data on elastic modulus and impact toughness.

6)    The term of corrosion resistance in this work is confusing. In the coating science, the corrosion is defined as the process of metal oxidation. In this work, the authors only studied the water permeation and chemical resistance.

7)    Figure 9: the scale bars are missing in the optical micrographs. The authors heavily discussed the wear resistance mechanism about the microcracking and polymer deformation based on figure 9. This reviewer could not find any microcracking and polymer deformation from figure 9. In addition, Figure 9 is optical micrographs not electron micrographs.

8)    Conclusions: the authors only reported the summary on experimental data but missed statistical calculations.

Author Response

It is corrected in the text of the publication

Reviewer 2 Report

The manuscript cannot meet the requirement of Polymers. It is simple to draw some figures with response surface methods. It is not investigated the materials system. In addition, the experimental data and detail is missing. The detailed comments are as follows:

1. The references are out of data. I cannot find newly reference. Please cite more newly papers in the introduction part.

2. Line 23-25, the authors claimed that " it allows getting of composites with improved mechanical and thermophysical properties due to the interaction of the active C-O and OH-groups on the surface of particles with macromolecules and segments of the epoxy binder, which will significantly increase the lifetime of coverings of transport parts“. The authors lack evidence to support this conclusion. Please offer some characterization data.

3. Please offer the experimental detail of the samples. In addition, please add photographs of the tested samples.

Author Response

(The authors gave the same response as above.)

Reviewer 3 Report

Dear,

The authors developed functional polymeric nanocomposites for restoring transport vehicles. The manuscript needs some minor revisions:

> Please present the main finding of the manuscript in the abstract;

> Please, authors should reduce the size of the introduction. Present only the important aspects;

> Page 5. Please specify the type of elastic modulus (tensile and flexural). Add the test speed, the load used, the machine used;

> Research methods. Please specify how many samples were tested on each property;

> Statistical Methods. Report the criteria adopted in the statistical test;

> Results and discussion. Please, in the numerical results, the authors must add the experimental errors;

> Why was the tensile or flexural strength properties added?

> 8 and 9 Figures. Add the name of Figure 8 to the "y" scale and the scale to the morphology of Figure 9;

Moderate editing of English language required

Author Response

(The authors gave the same response as above.)
